# New Bioretention Drainage Channel as One of the Low-Impact Development Solutions: A Case Study from Poland

Agnieszka Stec *  and Daniel Słyś 

Department of Infrastructure and Water Management, Rzeszow University of Technology, Al. Powstańców Warszawy 6, 35-959 Rzeszow, Poland; daniels@prz.edu.pl
* Correspondence: stec_aga@prz.edu.pl

**Abstract:** In recent years, as a result of intensive urbanisation, a significant increase in the surface of impermeable areas has been observed, which results in changes in the hydrological cycle of catchments. In order to counteract these changes, low-impact development (LID) solutions are increasingly being implemented in urban catchments, including bioretention systems. Taking this into account, a new bioretention drainage channel (BRC) was designed, whose main task is retention, infiltration, and pre-treatment of rainwater. The pilot laboratory tests carried out on two BRC prototypes (K1 and K2) showed that the average rate of reduction of mineral-suspended solids from rainwater was 69% and 57%, respectively, for K1 and K2. Analysing the results of the research, it was found that the bioretention drainage channel is characterised by very high efficiency in removing petroleum hydrocarbons from rainwater, and the reduction rate of these pollutants for both the K1 and K2 channels was close to 100%. In turn, hydrodynamic studies carried out on the model of the urban catchment showed that the implementation of BRCs will reduce the peak runoff by more than 82%, and the maximum flow in the sewage network by 83%.

**Keywords:** rainwater; stormwater; sustainable development; bioretention; drainage channel; LID

## 1. Introduction

The changing climate and the increasing incidence of extreme precipitation events with alternating long-term droughts make the management of rainwater in urban areas a topic of increasing interest [1,2]. The expansion of cities and the related increase in the sealing of areas, and thus the amount of rainwater discharged into sewage systems, disturbs the natural circulation of water in nature, contributing to lowering the groundwater level, excessive drying and erosion of soils, and increasing pollution of rivers and lakes [3]. Traditional drainage systems that were designed years ago for different terrain and hydrological conditions are often unable to drain large amounts of rainwater in a short period of time, leading to the occurrence of so-called urban floods [4,5]. These events are often catastrophic in their consequences, causing not only economic but also social losses [6,7].

To counteract these adverse changes, it is necessary to implement the principles of sustainable rainwater management into practice [8]. The overriding idea of activities in this area is to retain rainwater at its source and to counteract the phenomena of sudden, intensive, and cumulative discharges of rainwater into surface receiving bodies. According to the Water Framework Directive of the European Community 2000/60/EC of 23 October 2000, which establishes a framework for action in the field of water policy, associated countries are obliged to rationally use and protect water resources, in accordance with the principle of sustainable development [9]. These activities should be understood as the activity of states in areas which in their strategy aim at: (1) meeting the demand for water of the population, agriculture, and industry, (2) promoting sustainable water use, (3) protecting waters and ecosystems in good ecological status, (4) improving water quality and the condition of ecosystems degraded by human activity, (5) reducing groundwater

pollution, and (6) reducing the effects of floods and droughts. Traditional drainage systems, in which rainwater is treated as sewage/waste/a problem that should be disposed of as soon as possible, do not meet the above assumptions and are perceived as unsustainable [10]. One of the sustainable approaches in rainwater management is low-impact development (LID) [11]. LID refers to techniques and principles that aim to reduce stormwater runoff into sewers, recharge groundwater, increase infiltration, and protect rivers and watercourses. Achieving these goals enables the use of solutions such as retention reservoirs [12,13], infiltration facilities and devices [14,15], and rainwater-harvesting systems [16,17].

Among the LID solutions, bioretention systems (BRS) are widely used, which not only reduce the runoff of rainwater, but also reduce the amounts of pollutants transported with these waters [18]. This is of great importance because rainwater, especially from urbanised areas, is characterised by significant amounts of pollutants, which often limit the possibility of their direct management [19–21]. Among the pollutants generated by human activities, those generated by traffic play a fundamental role. These pollutants runoff with rainwater from streets, car parks, and squares [22,23]. One of the most important parameters in assessing the degree of pollution of rainwater in urban catchments is the content of suspended solids, which accumulate environmentally harmful substances, including microplastics and heavy metals. As numerous studies have shown, the amount of pollutants in rainwater effluent varies and depends on many factors, including catchment land use, the degree of surface sealing, the intensity of vehicle traffic, the amount and duration of rainfall, and the frequency of rainfall [3,24,25].

BRS research in recent years has mainly focused on determining the benefits of their use, in terms of hydrology [26,27] as well as the pollutant removal capacity [28,29]. Several studies have shown that the use of bioretention facilities is an effective solution in reducing the negative environmental impacts of urbanisation. For example, Shreshta et al. reported that the average reduction in stormwater volumes in bioretention cells located along roads was 75% (range 48–96%), peak flows was 91% (range 86–96%), and total suspended solids (TSS) was 94% (range 89–96%) [30]. Additionally, research by Mahmoud et al. shows that bioretention cells can significantly reduce pollutants from rainwater, including TSS (94–100%) [31]. Bioretention systems can also be effective in removing heavy metals [32], nitrogen and phosphorus [33,34], and bacteria [35].

A number of researchers have shown that bioretention systems are also an effective solution for reducing rainwater runoff from catchment areas. Greksa et al. [36] conducted BRS simulation studies for four sites and obtained the total average volume reduction, ranging from about 43% to 94%. On the other hand, researchers from Brazil obtained a 70% runoff reduction capacity for the case in which treated rainwater in a bioretention cell was directed to a reservoir, from which the water was a source for non-potable uses [37].

Despite that research shows that rainwater is a valuable resource, in many countries, including Poland, it is most often perceived as waste that should be disposed of as soon as possible. For this reason, the large-scale implementation of sustainable rainwater management systems, including LID facilities, is difficult, and the lack of clear and coherent legal provisions in this area additionally exacerbates the problem [38]. In Poland, only a few cities have a combined sewage system, the rest are equipped with separate or mixed systems. If the wastewater is discharged directly to the receiver, it may result in the discharge of significant amounts of pollutants into the waters [39]. Therefore, research was undertaken to determine the effectiveness of removing pollutants from rainwater on an innovative bioretention drainage channel, the construction of which is an original solution [40]. Laboratory pilot tests were carried out for two prototypes of this solution. A hydrodynamic model of a real urban catchment in which the devices under study were implemented was also developed. This allowed simulation studies to be carried out to analyse the impact of the new drainage channel solution on rainwater runoff from the catchment.

## 2. Materials and Methods

### 2.1. Research Stages

The research presented in this paper was carried out according to the procedure shown in Figure 1. Firstly, the state-of-the-art in the field of linear facilities and devices for drainage of rainwater from catchment areas was analysed. On this basis, a concept for a new drainage device was developed. In addition, technical and design documentation, a 3D model of the device, and an application to the patent office were prepared. In the next stage, pilot laboratory tests were carried out on two prototypes of the bioretention drainage channel (BRC). These tests were carried out as part of a research project carried out under a grant programme co-financed by the European Regional Development Fund as part of Priority Axis No. I, 'Competitive and innovative economy'. Subsequently, a hydrodynamic model of a real urban catchment was made, in which the BRC system was implemented.

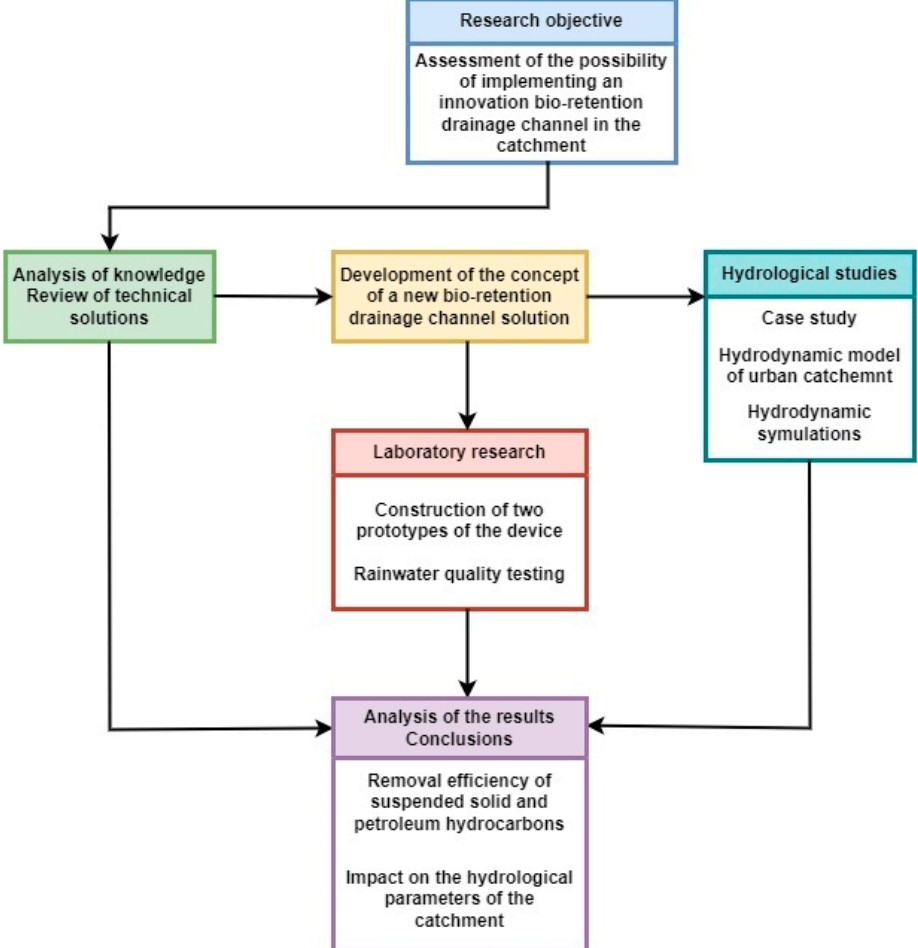

**Figure 1.** Research plan.

### 2.2. Characteristics of a New Bioretention Drainage Channel Solution

Taking into account the most frequently used linear drainage in practice, a bioretention drainage channel (BRC) was developed for temporary retention, drainage, and treatment of rainwater [40]. The implementation of BRCs in urban catchments will increase biodiversity, improve the microclimate through retention and evaporation, and above all, increase the inflow of water to the ground. The developed device will allow reducing the amount of rainwater discharged into the underground sewage system and will create the possibility of obtaining additional green areas, which are lacking in urbanised areas. The BRC is an innovative device, the construction and operation of which is an alternative to similar, linear rainwater drainage devices. Traditional solutions usually allow rainwater to be

discharged to the sewage system or directly to the receiver, often causing pollution. The use of a multi-layer filter-insert and vegetation with phytoremediation capabilities will increase its efficiency and reduce the negative impact on the environment.

The construction of the bioretention drainage channel without the filter-insert filling is shown in Figure 2. As shown in the figure, the bioretention drainage channel is made of an outer casing (1), which contains four side walls (2) and a connecting bottom wall (3), in which there are drainage holes (4). In the inner space of the channel, a filter-insert (5) is hung, containing four vertical walls (6) and a bottom wall (7) connecting them. To the vertical walls of the filter-insert, handles are fixed (8). On these handles, the filter-insert is hung. In the bottom of the filter-insert, there are flow holes (9), through which rainwater flows. Inside the BRC, there is an accumulation space (10) that allows rainwater to be temporarily retained.

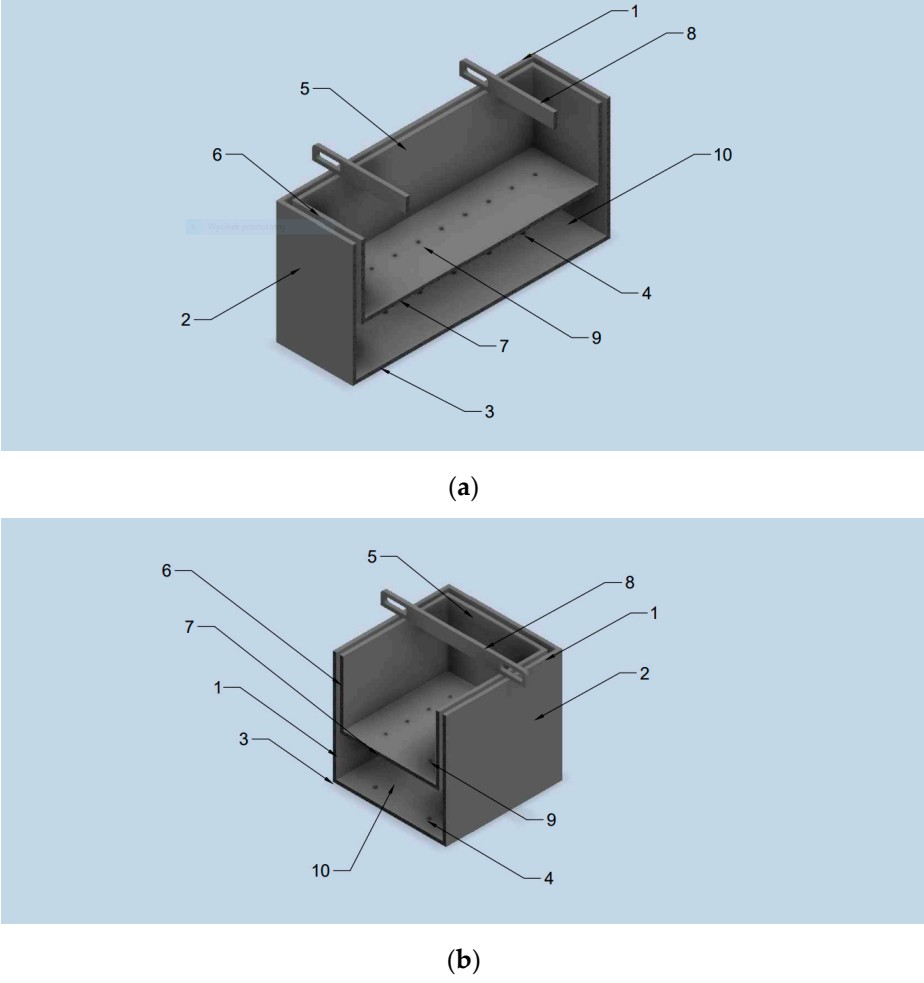

**Figure 2.** The 3D model of the construction of the bioretention drainage channel module: (**a**) longitudinal cross-section, and (**b**) cross-section: 1—external construction (casing), 2—side walls, 3—bottom wall, 4—outflow holes, 5—filter-insert, 6—vertical walls, 7—bottom wall, 8—filter-insert handles, 9—flow holes, 10—accumulation space.

An integral part of the bioretention drainage channel is the filter-insert, the structure of which is shown in Figure 3. Inside the filter-insert, a drainage layer (11) is placed over the entire bottom surface, on which a non-woven filtering layer (12) is placed. Subsequent layers are the chemically active layer (13) and the layer of vegetation substrate with plants (14). The filtration layer prevents substrate particles from entering the drainage layer. The bottom of the filter-insert is also lined with a filter layer (15), in the form of a

non-woven fabric, whose task is to prevent the drainage layer particles from penetrating through the flow holes.

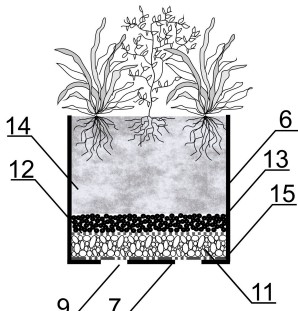

**Figure 3.** Construction of the filter-insert: 6—vertical walls, 7—bottom wall, 9—flow holes, 11—drainage layer, 12—filter non-woven fabric, 13—chemically active layer, 14—vegetation substrate with plants, 15—non-woven fabric filter.

*2.3. A prototype of a Bioretention Drainage Channel*

In order to carry out the laboratory tests, it was necessary to make two prototypes of the developed innovative bioretention drainage channel. Due to the technical aspects, the physical models of the tested device were designed from PEHD material, ensuring the appropriate strength of the structure. Structural elements with the following dimensions were designed:

- Filter-insert: 30 cm × 30 cm × 100 cm, the diameter of the holes in the bottom is 15 mm.
- Outer housing: 35 cm × 50 cm × 105 cm, the diameter of holes in the bottom is 20 mm.

The choice of material for the vegetation layer was made based on the available range of products on the market. A soil substrate was used, which is a specially developed mineral-organic mixture. This material ensures:

- Stable and long-lasting plant vegetation,
- Water for the proper development of the plants,
- Drainage of excess water into the drainage layer,
- Proper aeration of plant roots,
- Resistance to subsidence (mineralisation),
- Resistance to weather conditions (frost, wind),
- Optimal content of organic components necessary for proper plant growth.

In order to create a chemically active biodegradable vegetation layer, a layer of activated carbon produced from coconut shells was made under the substrate layer. This carbon is characterised by its high porosity, and therefore has a very high absorption capacity, allowing it to retain contaminants on the surface of its particles.

The drainage layer was made of expanded clay, which is a natural granulate formed during the clay-firing process. It is ideally suited for the preparation of a specialised substrate for growing plants. Medium and coarse fraction-expanded clay (8–16 mm) is suitable for creating drainage layers for plants grown in large pots. By creating a drainage layer of expanded clay at the bottom of the filter-insert, water can be drained away in good time and plants can be prevented from overflowing and rotting.

The bioretention drainage channel will function in changing weather conditions. Therefore, the selection of plant species had to consider a number of features related to the possibility of their proper growth. Plants with an undeveloped root system, low nutritional and soil requirements, and low susceptibility to diseases and pests were selected. In addition to features related to habitat requirements, these species are characterised by high resistance to frost, water, and thermal stress, which is particularly important in the currently changing climate, and especially in the increasingly frequent long-term droughts.

Species selected for planting in the bio-corridor were *Pennisetum alo-pecuroides*, *Heuchera x hybrida*, *Echinacea,* and *Carex*.

*Pennisetum alopecuroides* is characterised by an interesting appearance and relatively small habitat requirements. This creates great opportunities to use this species for planting in BRCs, with particular emphasis on cities, where habitat conditions are extremely unfavourable for plants (e.g., air and soil pollution, high temperatures). The low requirements in relation to the substrate are conducive to the cultivation of this species. It is characterised by a compact and tufted habit. It grows up to a maximum of 1 m.

*Heuchera x hybrida* is a perennial belonging to the Saxifragaceae family, usually reaching up to 40 cm. It has short, heavily leafy shoots that form a low and compact clump. *Heuchera* is a very hardy perennial. This perennial can grow in one place for many years. It is resistant to almost all diseases and pests. It is characterised by very low nutritional requirements.

*Echinacea* is a perennial reaching a maximum of about a metre in height. At the base of the plant, a clump of broad, lanceolate leaves of dark green colour forms. From between them grow straight shoots covered with smaller leaves. It is a popular, very easy-to-grow garden perennial that has versatile uses. Echinacea blooms from July to October. When it comes to soil selection, it is not a demanding plant.

*Carex* is a species with low cultivation requirements, and its adaptability allows it to grow both in full sun and in the shade. They are suitable for cultivation in almost any soil condition. The usefulness of this group as ornamental plants is determined not only by their decorative qualities but also by the low habitat requirements, low susceptibility to diseases and pests, as well as the high adaptability to changing growing conditions.

### 2.4. Laboratory Tests of the Bioretention Drainage Channel

An important function of the bioretention drainage channel is the ability to pre-treat rainwater, which will then be infiltrated into the ground. According to Polish law, in order to discharge water into the ground or into other waters, it is important to meet conditions where the content of pollutants in excess of 100 mg/L of total suspended solids and 15 mg/L of petroleum hydrocarbons will not be exceeded [41]. Taking this into account, qualitative studies have focused attention on these parameters to assess the efficiency of water treatment in the developed unit.

Two prototypes of the device were installed in August 2022 in the Laboratory of Measurement and Control Techniques for Water and Wastewater Transport of the Department of Infrastructure and Water Management. They were subjected to testing in the period of September to October 2022. The bioretention drainage channels for the study were designated K1 and K2 (Figure 4). Eight seepage water (leachate) samples were taken from each prototype unit for qualitative testing. Due to the possibility of contaminants leaching out of the filter materials in the BRCs, the first two samples were blank. In this case, contaminant-free water was percolated through the K1 and K2 channels. Afterwards, prepared rainwater with averaged contaminant concentrations was percolated through the devices. This mapped the actual conditions under which the BRCs would operate and made it possible to determine the pollutant removal efficiency. Unleaded 95 octane petrol and clay dried for 24 h at 60 °C in a laboratory dryer were used to prepare the rainwater. Waters with a pollutant concentration of 200 mg/L of total suspended solids and 20 mg/L of petroleum hydrocarbons were percolated through the K1, and with a pollutant concentration of 400 mg/L of total suspended solids and 30 mg/L of petroleum hydrocarbons through K2.

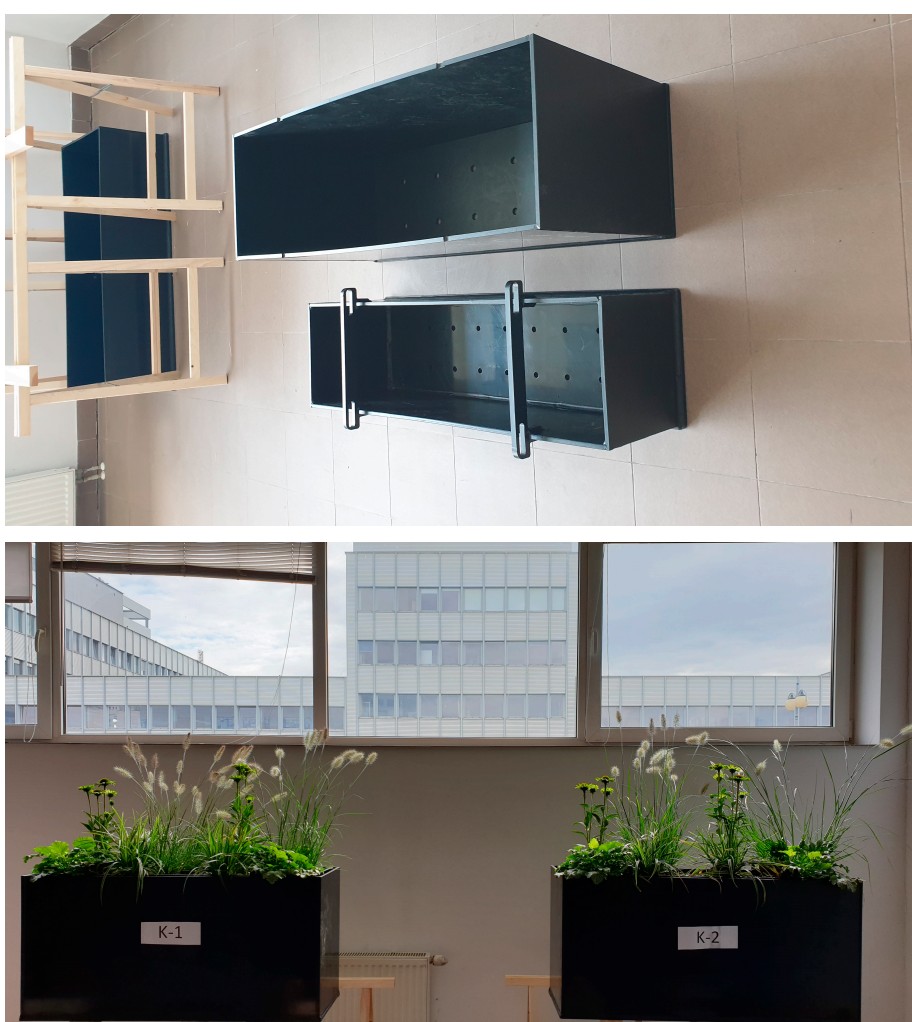

**Figure 4.** Bioretention drainage channel modules installed in the laboratory.

Qualitative testing was carried out by the accredited laboratory SGS Polska Sp. z o.o. Leachate samples collected by the research team were picked up on the same day by a laboratory employee. Total suspended solids were determined using the gravimetric method described in PN-EN 872:2007 + Ap1:2007 (A) [42]. This is a method for the determination of suspended solids in water, wastewater, and treated wastewater using filtration through glass fibre filters. The limit of quantification was 2 mg/L. The upper limit of quantification was not determined. In turn, the content of petroleum hydrocarbons in the samples was determined by gas chromatography in accordance with PN-EN ISO 9377-2:2003 (A) [43]. This is a gas chromatographic method for determining the mineral oil index in water, suitable for testing surface water, sewage, and treated sewage, allowing the mineral oil index to be determined for concentrations higher than 0.1 mg/L.

*2.5. Modelling a Bioretention Drainage Channel—Case Study*

A hydrodynamic model of a small urban catchment was made to test the impact of a bioretention drainage channel (BRC) on catchment runoff. The model was created in the Storm Water Management Model software developed by the U.S. Environmental Protection Agency, which allows LID facilities to be included in the catchment [44]. A residential area located in Rzeszów in south-eastern Poland was selected for the study (Figure 5). It is planned to build an additional car park next to the existing multi-family buildings. The surface of the car park, measuring 15 × 100 m, is designed with concrete paving blocks. Studies were carried out for two variants of the car park. The first variant involves draining

rainwater from the entire surface of the car park into the drainage network, while the second variant includes the construction of two linear drainage systems made of BRC modules (Figure 5).

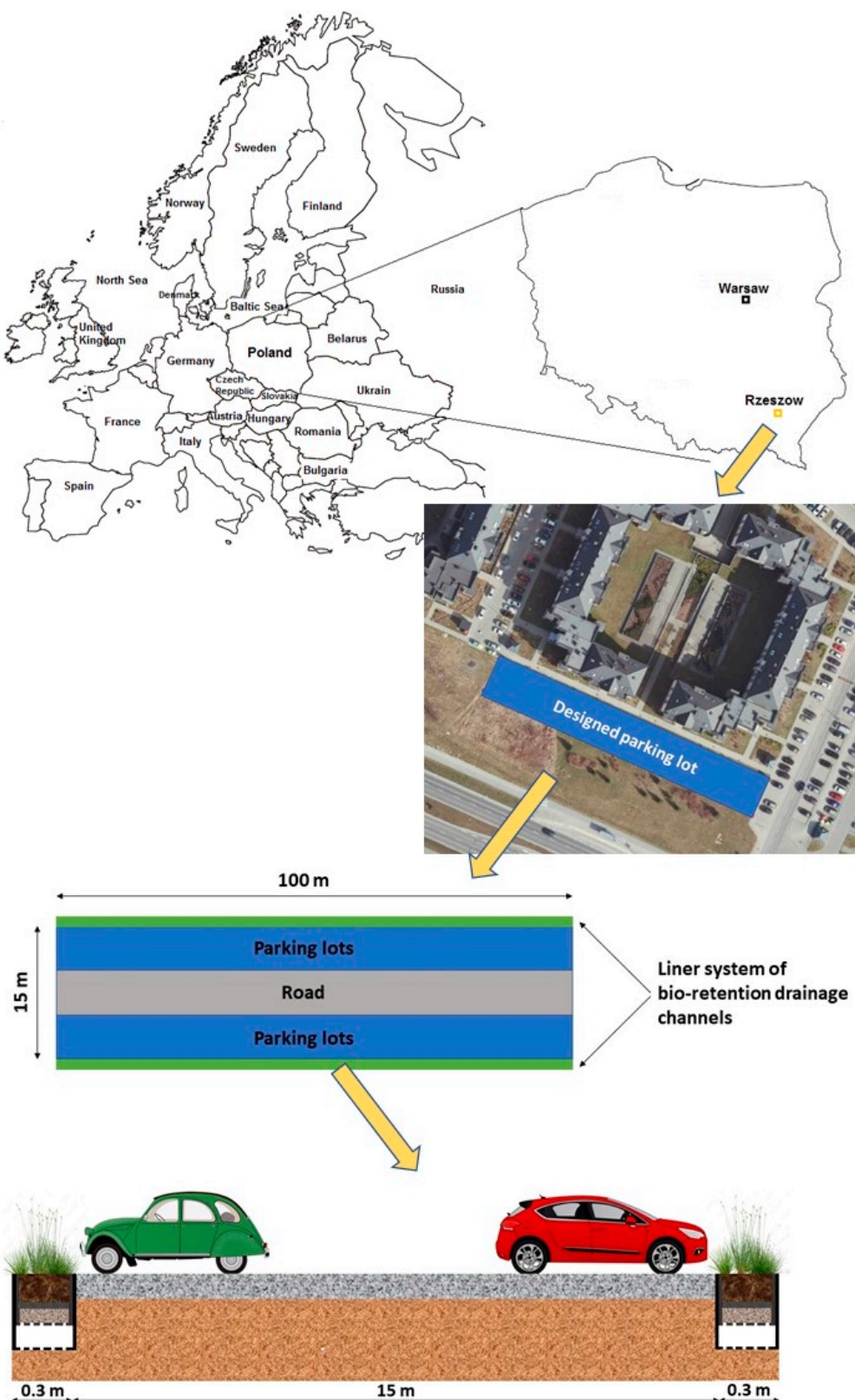

**Figure 5.** Localisation of the case study.

In the SWMM model, the newly developed bioretention drainage channel was used as a bioretention cell with modifications in terms of the layer layout. This object in the software functions as shown in Figure 6. In the bioretention cell, it is possible to include several horizontal construction layers. The surface layer receives both direct rainfall and runoff directed at it from other areas. Water then infiltrates into the soil layer below and is also partially subject to evapotranspiration (ET). The soil layer is a modified mixture that supports plant growth. The lowest layer is used to store water and further infiltrate into the native soil. It usually consists of crushed stone or gravel. It also allows water to drain through a pipe system if designed [45]. Considering the design of the bioretention cell and its functions, it was considered the most suitable of the LIDs available in SWMM to be modelled in the catchment of the developed solution. The catchment and bioretention drainage channel characterisation data used in the hydrodynamic model of the catchment are shown in Tables 1 and 2. The input data were adopted from recommendations by the U.S. Environmental Protection Agency [45] and other studies [46,47]. In the SWMM programme, precipitation is transformed into an effective runoff, determined as the runoff from a linear basin whose filling is equal to the amount of water that fell on a given surface after taking into account losses due to evaporation, soaking, and water retention in depressions in the land. Simulations were carried out using real rainfall data from the period 2007–2008.

**Table 1.** Input data for the hydrodynamic model of the analysed part of the catchment.

| Parameter | Value |
|---|---|
| Land surface slope, % | 2 |
| Manning's coefficient for impervious surfaces | 0.013 |
| Manning's coefficient for pervious surfaces | 0.15 |
| Impervious depression storage, mm | 2 |
| Pervious depression storage, mm | 4 |
| Percent imperviousness, % | 70 |

**Table 2.** Values of parameters characterising the bioretention drainage channel (BRC) implemented in the hydrodynamic model of the analysed catchment.

| Parameter | Value |
|---|---|
| *Surface layer* | |
| Berm height (mm) | 250 |
| Vegetation volume | 0.2 |
| Surface roughness | 0.13 |
| Surface slope (%) | 1 |
| *Soil layer* | |
| Soil thickness (mm) | 300 |
| Hydraulic conductivity (mm/h) | 250 |
| Suction head (mm) | 50 |
| Porosity | 0.6 |
| Field capacity | 0.5 |
| Wilting point | 0.2 |
| Conductivity slope | 44 |
| *Storage layer* | |
| Thickness (mm) | 50 |
| Void ratio (voids/solids) | 0.3 |
| Seepage rate (mm/h) | 28 |
| Clogging factor | 0 |

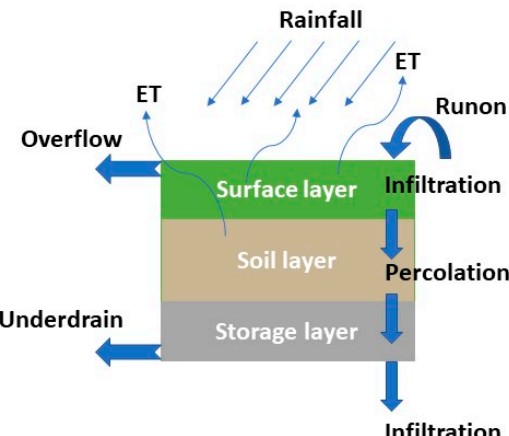

**Figure 6.** Functional diagram of the bioretention cell in SWMM software (based on [45]).

## 3. Results and Discussion

### 3.1. Laboratory Research of Bioretention Drainage Channels

Preliminary pilot tests carried out on two prototypes of the bioretention drainage channel made it possible to estimate the efficiency of removal of the analysed pollutants from rainwater. The results of the research are summarised in Table 3.

**Table 3.** Results of qualitative analyses for water leachates from the tested prototypes of the K1 and K2 channels.

| Sample | Bioretention Drainage Channel K1 | | Bioretention Drainage Channel K2 | |
|---|---|---|---|---|
| | Total Suspended Solids Concentration ZK1, mg/L | Concentration of Petroleum Hydrocarbons WK1, mg/L | Total Suspended Solids Concentration ZK2, mg/L | Concentration of Petroleum Hydrocarbons WK2, mg/L |
| 1 * | 102 | <0.10 | 105 | <0.10 |
| 2 * | 129 | <0.10 | 186 | <0.10 |
| 3 | 168 | <0.10 | 293 | <0.10 |
| 4 | 139 | <0.10 | 211 | <0.10 |
| 5 | 175 | <0.10 | 346 | <0.10 |
| 6 | 274 | <0.10 | 336 | <0.10 |
| 7 | 163 | <0.10 | 382 | <0.10 |
| 8 | 143 | <0.10 | 341 | <0.10 |

* Blank sample.

The final total suspended solids concentrations in the rainwater (leachate) in samples 3 to 8 were calculated from Equations (1) and (2) after taking into account the average pollutant concentration in the blank samples. The test results are shown in Table 4.

$$CZK1 = ZK1 - (102 + 129)/2, \tag{1}$$

$$CZK2 = ZK2 - (105 + 186)/2, \tag{2}$$

where: CZK1—the final total suspended solids concentration in samples from the bioretention drainage channel K1, mg/L, CZK2—the final total suspended solids concentration in samples from the bioretention drainage channel K2, mg/L, ZK1—the total suspended solids concentration in samples from the bioretention drainage channel K1, mg/L, and ZK2—the total suspended solids concentration in samples from the bioretention drainage channel K2, mg/L.

**Table 4.** The final results of qualitative tests for water leachates from the K1 and K2 prototypes.

| Sample | Bioretention Drainage Channel K1 | | Bioretention Drainage Channel K2 | |
| | Total Suspended Solids Concentration CZK1, mg/L | Concentration of Petroleum Hydrocarbons WK1, mg/L | Total Suspended Solids Concentration CZK2, mg/L | Concentration of Petroleum Hydrocarbons WK2, mg/L |
|---|---|---|---|---|
| 3 | 52.5 | <0.10 | 147.5 | <0.10 |
| 4 | 23.5 | <0.10 | 65.5 | <0.10 |
| 5 | 59.5 | <0.10 | 200.5 | <0.10 |
| 6 | 158.5 | <0.10 | 190.5 | <0.10 |
| 7 | 47.5 | <0.10 | 236.5 | <0.10 |
| 8 | 27.5 | <0.10 | 195.5 | <0.10 |

In the case of petroleum hydrocarbons, test results preceded by (<) indicate a result outside the lower measuring range of the method, where the value provided is the lower limit of quantification with the corresponding uncertainty.

The concentrations of contaminants obtained in the rainwater (leachate) made it possible to determine the degree of reduction of the contaminants in relation to the concentrations in the treated rainwater. By using a filter-insert with appropriately sized layers and vegetation, it was possible to retain a significant amount of contaminants in the bioretention drainage channels. Table 5 shows the level of reduction of total suspended solids and petroleum hydrocarbons for the two tested prototypes of the BRC. Analysing the results of the tests carried out, it was found that the bioretention drainage channel, which is the subject of the research project, had a very high efficiency in removing petroleum hydrocarbons from rainwater, and the reduction rate of these pollutants in both the K1 and K2 prototype was close to 100%. In the qualitative analyses of the leachates, the concentration of this pollutant was below the level of quantification. It is also significant that the concentration of petroleum hydrocarbons did not increase with the duration of the tests, which could suggest their leaching from the filter media layers. Such high efficiency of the removal of petroleum substances in the tested device is a significant advantage over traditional drainage channels, as petroleum hydrocarbons have strong toxic and carcinogenic properties, easily enter the environment, causing contamination, and above all, are dangerous to human health and life. The implementation of the bioretention drainage channels into engineering practice and their implementation in real catchments will, therefore, contribute to reducing the negative impact of human activities on the environment.

**Table 5.** The reduction of pollutants in leachate from the tested prototypes of the bioretention drainage channel.

| Sample | Bioretention Drainage Channel K1 | | Bioretention Drainage Channel K2 | |
| | Total Suspended Solids, % | Petroleum Hydrocarbons, % | Total Suspended Solids, % | Petroleum Hydrocarbons, % |
|---|---|---|---|---|
| 3 | 74 | ~100 | 63 | ~100 |
| 4 | 88 | ~100 | 84 | ~100 |
| 5 | 70 | ~100 | 50 | ~100 |
| 6 | 21 | ~100 | 52 | ~100 |
| 7 | 76 | ~100 | 41 | ~100 |
| 8 | 86 | ~100 | 51 | ~100 |

Taking into account the reduction in the concentration of total suspended solids in the K1 prototype, it can be considered to be very high, ranging from 70% to 88% (ignoring sample 6, which significantly deviated from the other results—outlier). Considering the outlier values, for all six samples, the average degree of reduction of total suspended

solids in the leachate from K1 was 69%. It should be noted that, except for sample 6, the concentration of total suspended solids was well-below 100 mg/L, which is below the permissible concentration for rainwater discharged to water and ground, as defined in the current legislation. In the case of the K2, the concentration of total suspended solids in the leachate, except for sample 2 (outlier), was above the permissible value of 100 mg/L. The level of reduction in the concentration of this pollutant ranged from 41% to 63% (excluding sample 2, which was significantly different from the other concentrations of total suspended solids for the K2 channel), while the average level for all samples taken was 57%. The higher concentration of total suspended solids in the leachate from the K2 bioretention drainage channel may be caused by the high concentration of these contaminants in the treated rainwater flowing through the filter-insert, as well as by contaminants leaching from the soil material of the filter media, which, due to the short study period, may not yet have been worked in.

During the realisation of the study, observations were conducted over two months to assess the viability of the planted vegetation. It is noteworthy that the vegetation layer was exposed to significant stress due to the amounts of petroleum pollutants dosed. After two months of testing, the plants were found to be in at least good condition and, except for a few dried leaves, no plant rots were observed. It can, therefore, be concluded that the plant species and the type of soil material constituting the vegetation layer were correctly selected. It is worth emphasising, however, that for a full assessment of the functioning of the vegetation layer, further tests under real conditions should be carried out, at least on an annual basis, covering all stages of the vegetation life, as well as underwater and in cold stress conditions.

There are no solutions that are identical to the developed bioretention drainage channel, which limits the possibility of comparing the results obtained with other studies. Producers of drainage channels, which in addition to discharging rainwater also have pre-treatment capabilities, provide data mainly on the reduction of heavy metals. Research results on the removal of total suspended solids (TSS) in various types of LID facilities are available in the literature. For example, Nazarpour et al. [48] notes that the average TSS reduction rate in a bioretention cell exceeds 78%. In turn, in [49], the level of TSS reduction for a similar device ranged from 79% to 97%.

### 3.2. Hydrodynamic Research of BRC

The simulation studies carried out have shown that the implementation of innovative bioretention drainage channels (BRCs) in a real urban catchment as one of the LID solutions can have a significant impact on both the hydrological parameters of the catchment and the sewerage system.

Analysing the change of hydrological conditions in the considered catchment, it was noted that the runoff from the catchment in which BRCs were implemented was reduced by more than 82% (peak runoff). The results of the study for selected rainfall events are shown in Figure 7. The level of peak runoff reduction is influenced, among other things, by the type of soil in which the water infiltration facilities were implemented. Winston et al. [50] conducted a study for bioretention cells located in clay soils and obtained a 56% maximum peak runoff reduction. In turn, Kandel et al. [51] presented a study for a car park from which runoff was directed to bioretention cells. Their application resulted in a volume reduction of 73%. Similar studies for car parks were described in [31,52], where the reduction in rainwater runoff was 98% and 82%, respectively.

The application of the tested drainage channels in the catchment also resulted in an increase in total infiltration of up to 85% compared to the car park variant without BRCs. The dependence of the infiltration of water into the ground on rainfall is shown in Figure 8. Avellaneda et al. [53] created a hydrodynamic model of a small residential catchment in SWMM, in which they included LID facilities such as bioretention cells. The results showed that there was an increase in infiltration of only 7.6%.

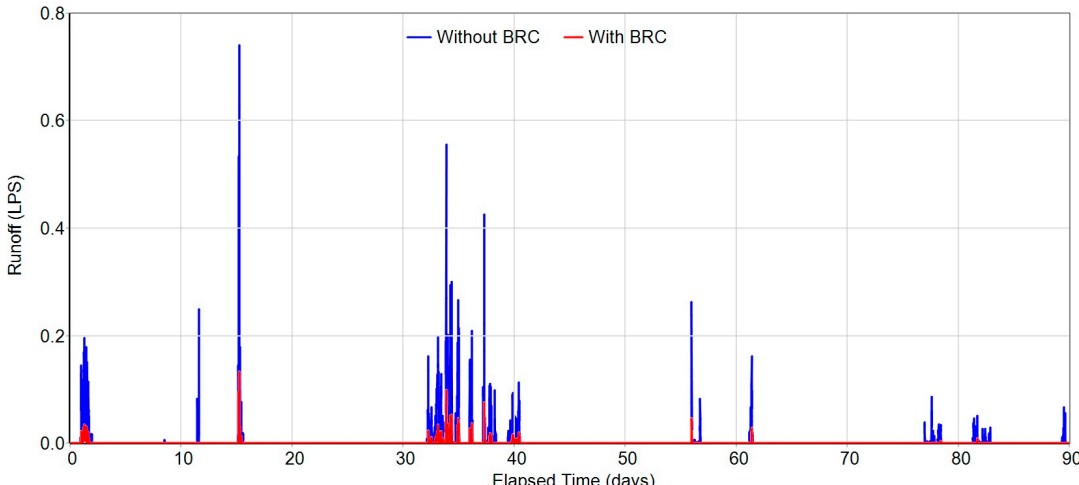

**Figure 7.** Rainwater runoff from the analysed car park for selected 2007–2008 rainfalls for variants of the car park with and without bioretention drainage channels (BRCs).

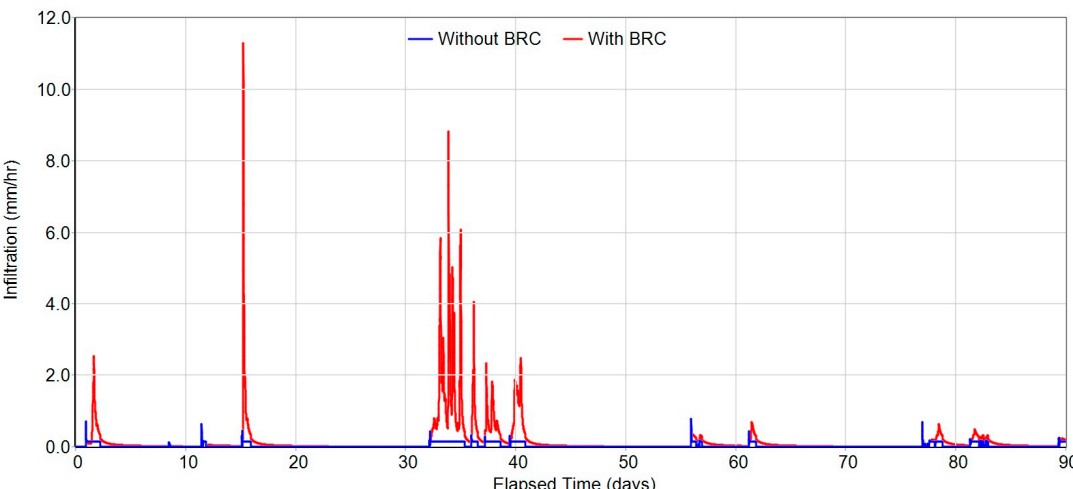

**Figure 8.** Infiltration of rainwater for selected 2007–2008 rainfalls for variants of the car park with and without bioretention drainage channels (BRCs).

This study also analysed the impact of the BRCs on the operation of the sewer system. According to the design, the planned car park will be connected to the sewer system. In the first variant, rainwater from the paved surface of the car park will be discharged into a circular channel with a diameter of 200 mm. In the second variant, only the excess water will runoff into the sewer system. Simulation results showing the flow of rainwater in the channel are shown in Figure 9. Implementation in the catchment area of the tested bioretention channels resulted in a reduction of maximum flow by approximately 83%. Wałęga et al. [54] conducted a simulation study for an urban catchment from which rainwater is discharged into a combined sewage system. They found that the application of a bioretention system in the analysed catchment would result in a reduction of the cumulative flow rates by almost 56% and of the flood wave volume by over 54%.

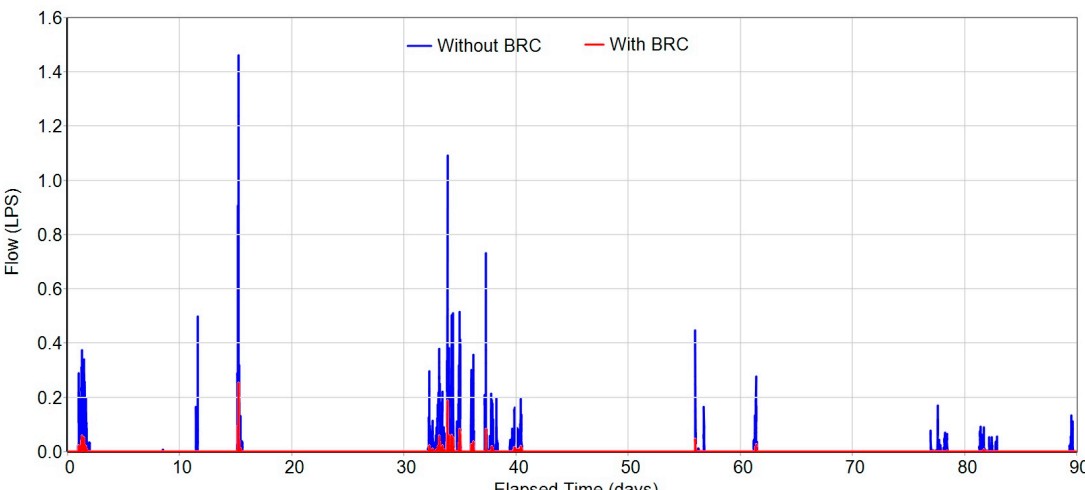

**Figure 9.** Rainwater flow in the channel connecting the car park to the sewer system for selected rainfalls from 2007 to 2008 for variants of the car park with and without bioretention drainage channels (BRCs).

## 4. Conclusions

The use of LID facilities in urban catchments offers many benefits, not only from a technical point of view but also for the lives of residents. The main disadvantage of most traditional drainage channels is the inability to infiltrate rainwater into the ground and the lack of vegetation, which not only has aesthetic value but also the ability to treat this water before further management. Taking this into account, a bioretention drainage channel (BRC) was developed for the temporary retention, drainage, and treatment of rainwater. The implementation of these channels in urban catchments will increase biodiversity, improve microclimates, and above all, increase groundwater recharge. The newly developed device will also allow the creation of additional green areas, which are particularly valuable in urban areas.

The laboratory and simulation studies have led to the following main conclusions:

- The designed construction of the BRC fulfilled its intended primary function, which is the pre-treatment of rainwater.
- The BRC was characterised by very high efficiency in removing environmentally and humanly harmful petroleum hydrocarbons from rainwater, where the degree of reduction of these pollutants was almost 100%.
- The degree of reduction in the concentration of total suspended solids was also high, averaging 69% and 57% for prototypes K1 and K2, respectively.
- The modular design of the bioretention drainage channel makes it possible to create structures of any size to suit local terrain conditions and rainwater volumes.
- The designed channel is a compact device with a smaller surface area than other objects with similar functions, such as infiltration basins.
- Plants such as *Pennisetum alopecuroides*, *Heuchera x hybrida*, *Echinacea,* and *Carex* are suitable species for planting in the bioretention drainage channel.
- The BRCs were characterised by very high retention and infiltration efficiencies, and their implementation in the catchment improved the hydrological parameters of the catchment.
- The implementation of the BRCs in the catchment made it possible to achieve a high degree of reduction of flows in the sewerage network (83%). It can be important in cases of connecting new catchments to the sewerage systems, which are often hydraulically overloaded in their current state.

The research results discussed in this article are from a preliminary analysis of the effectiveness of removing pollutants from rainwater on new bioretention drainage channels. They also form the basis for further research, which will be carried out by the research team

in real conditions. Prototypes of the device will be implemented in the urban catchment along the car park. The research will be conducted over a longer period of time, which will allow a thorough assessment of the impact of the variability of climatic conditions on the functioning of this solution.

**Author Contributions:** Conceptualisation, A.S. and D.S.; methodology, A.S. and D.S.; software, A.S.; validation, A.S. and D.S.; formal analysis, A.S. and D.S.; investigation, A.S. and D.S.; resources, A.S. and D.S.; data curation, A.S. and D.S.; writing—original draft preparation, A.S.; writing—review and editing, A.S. and D.S.; visualisation, A.S.; supervision, D.S.; project administration, A.S. and D.S. All authors have read and agreed to the published version of the manuscript.

**Funding:** This research was funded by Podkarpackie Centrum Innowacji (PCI), grant number N3_597, grant title "Research on an innovative drainage bio-channel".

**Data Availability Statement:** Research data available on request.

**Conflicts of Interest:** The authors declare no conflict of interest.

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
