# Peer review of "New Bioretention Drainage Channel as One of the Low-Impact Development Solutions: A Case Study from Poland"

_resources, doi:10.3390/resources12070082_

Round 1

Reviewer 1 Report

The paper of the authors from Poland is concerned with " New bio-retention drainage channel as one of the low impact development solution ”.

Thank you for the opportunity to review such interesting, useful, and actual paper.

The manuscript presents reviewing the state of the art of low-impact development solutions for being implemented in urban catchments, including bioretention systems.

This submitted article could be with the aim and scope of the MDPI journal Resources; and its Special Issue: Alternative Water and Energy Systems in the Buildings https://www.mdpi.com/journal/resources/special_issues/3ER89612S2 

• Abstract & introduction: These two parts are focused on the paper's main aim and the new contributions of authors to the state of the art. The abstract with keywords very effectively summarizes the manuscript.

The key objective for the authors is to show the hydrodynamic studies carried out on the model of the urban catchment in the specific condition of Poland, the city of Rzeszow.

• Materials & methods: Based on the very good basement of choosen research plan, the authors showed the topic from different points of view.

Their main objective was designing of a new bio-retention drainage 3D channel, whose main task is retention, infiltration and pre-treatment of rainwater. The pilot laboratory tests carried out on two BRC prototypes followed simulation in the Storm water management model software.  

• Results & discussion: The data are well-presented with relevant and current tables, figures, and references. Authors analyzed the results and they found that the bio-retention drainage channel is characterized by very high efficiency in removing petroleum hydrocarbons from rainwater, and the reduction rate of these pollutants for both the prototypes was close to 100%. In turn, hydrodynamic studies carried out on the model of the urban catchment showed that the implementation of BRCs in it will reduce the peak runoff by more than 82%, and the maximum flow in the sewage network by 83%..

With some adaptation, this paper and its conclusions could be use more generally by other researchers and countries.

Authors know about limitation their study and indicate to the future of their research.

I have only following comment for improving of manuscript before publishing.

Line 300: word „gdzie“ should be translate in english

Author Response

Responses to the Reviewer’s #1 comments:

The authors thank the Reviewer #1 for taking the time to evaluate the manuscript and for positive opinion. Thank you for your remarks on the subject relevance and the research carried out.

Comment: Line 300: word „gdzie“ should be translate in English

Response: The word „gdzie” was transleted to „where” (Line 300).

Reviewer 2 Report

This paper discusses innovative bio-retention drainage channels for temporary detention, drainage and treatment of rainwater. The implementation of these channels in urban catchments will increase biodiversity, improve the microclimate and, above all, increase groundwater recharge. The newly developed device will also allow the creation of additional green areas, which are particularly valuable in urban areas.

The article can be accepted in its current form, after taking into account two minor comments:

1.        Provide English-translated names of sources in the literature list.

2.        Arrange sources chronologically when citing them in groups.

Author Response

Responses to the Reviewer’s #2 comments:

The authors thank the Reviewer #2 for taking the time to evaluate the manuscript and for positive opinion. Considering his comments, some parts have been rewritten to clarify the manuscript.

Comment: Provide English-translated names of sources in the literature list.

Response: We thank Reviewer #2 for these insightful comments. The English-translated names of sources in the References were added (Lines: 549, 552-554, 556, 558-559).

Comment: Arrange sources chronologically when citing them in groups.

Response: Thank you for this comment. It was changed in the text and marked in red.
